# Warm Air Entrainment and Advection during Alpine Blowing Snow Events

**Nikolas O. Aksamit[1†], John W. Pomeroy[1]**

1 Centre for Hydrology, University of Saskatchewan, Saskatoon, Canada
† Current affiliation Institute for Mechanical Systems, ETH Zürich, Switzerland

*Corresponding author*: Nikolas Aksamit (naksamit@ethz.ch)

**Abstract.** Blowing snow transport has considerable impact on the hydrological cycle in alpine regions both through the redistribution of the seasonal snowpack and through sublimation back into the atmosphere. Alpine energy and mass balances are typically modelled with time-averaged approximations of sensible and latent heat fluxes. This oversimplifies non-stationary turbulent mixing in complex terrain and may overlook important exchange processes for hydrometeorological prediction. To determine if specific turbulent motions are responsible for warm and dry air advection during blowing snow events, quadrant analysis and Variable Interval Time Averaging was used to investigate turbulent time series from the Fortress Mountain Snow Laboratory alpine study site in the Canadian Rockies, Alberta, Canada during the winter of 2015-2016. By analyzing wind velocity and sonic temperature time series with concurrent blowing snow, such turbulent motions were found to supply substantial sensible heat to near surface wind flows. These motions were responsible for temperature fluctuations of up to 1°C, a considerable change for energy balance estimation. A simple scaling relationship was derived that related the frequency of dominant downdraft and updraft events to their duration and local variance. This allows the first parameterization of entrained or advected energy for time-averaged representations of blowing snow sublimation and suggests that advection can strongly reduce thermodynamic feedbacks between blowing snow sublimation and the near-surface atmosphere. The downdraft and updraft scaling relationship described herein provides a significant step towards a more physically-based blowing snow sublimation model with more realistic mixing of atmospheric heat. Additionally, calculations of return frequencies and event durations provide a field-measurement context for recent findings of non-stationarity impacts on sublimation rates.

## 1 Introduction

At least 40% of the world's population relies on the seasonal snowpack as a temporary reservoir of winter snowfall that then provides meltwater in spring and summer for downstream water use [Meehl et al., 2007]. However, after snow has fallen, it is often subjected to sublimation while at rest or amplified in-transit sublimation during redistribution. Blowing snow redistribution can result in vast amounts of frozen water moving between basins or, in the case of sublimation, being removed entirely from the surface water budget in wind swept regions. Blowing snow particles are highly susceptible to sublimation because of their high curvature, large surface area to mass ratio, and high ventilation rates [Dyunin, 1959; Schmidt, 1982].

While estimates may vary with climate, in the Canadian Rockies, blowing snow transport has been found to be responsible for sublimating up to 20% of the yearly snowfall [MacDonald et al., 2010].

Snow sublimation is typically studied at large temporal and spatial scales within hydrometeorological modeling frameworks because of the complexity of the processes and the difficulty of particle transport tracking [e.g. Pomeroy et al., 1993; Pomeroy and Essery, 1999; Déry and Yau, 2002; Lenaerts et al., 2012; Groot Zwaaftink et al. 2013; Musselman et al., 2015]. To accurately calculate all contributions to boundary layer energy balances, latent heat flux estimates rely on an accurate sublimation model, and a precise understanding of how much energy is available for snow particle phase change. While latent

and sensible heat exchanges between the turbulent atmosphere and snow particles can be represented by a system of coupled partial differential equations, they require forcing terms for dry-air entrainment and horizontal advection that are still poorly understood and have not been based on observed physical mechanisms, if they are included at all [Bintanja, 2001]. The decrease in temperature and increase in humidity in the atmosphere caused by snow sublimation may play a crucial limiting role in snow sublimation, but many blowing snow models struggle to capture the process of this feedback, which can result in

unrealistic atmospheric conditions in near-surface boundary layers and subsequent errors in calculations of the blowing snow sublimation rate [Pomeroy and Li, 2000, Dery and Yau, 1999, 2001; Groot Zwaaftink et al. 2013; Musselman et al, 2015]. Numerical investigations of snow sublimation from a numerical modeling approach have recently provided new insights into non-steady state aspects of sublimation [Dai and Huang, 2014; Li et al., 2017; Sharma et al., 2018] and the efficacy of the nearly-universally used Thorpe and Mason [1966] model [see Schmidt, 1972] at high temporal and spatial resolution [Sharma

et al., 2018]. Extending non-stationary sublimation models to alpine and other complex terrain environments could lead to reduced uncertainty in blowing snow sublimation models. Little research has been conducted to better understand the energy available for snow sublimation from entrainment or advection processes in natural atmospheric turbulence or the influence that resultant air temperature fluctuations may exert on sublimation rates. Recently, Grazioli et al. [2017] found that over long timescales, persisent katabatic winds in Antarctica provide a consistent supply of unsaturated air that can contribute to

significant snow sublimation. In East Antarctica, they calculated up to 35% of total yearly snowfall can be lost in this manner. The objective of this research is to investigate turbulent structures down to sub-second timescales and identify their synchronization with near-surface temperature fluctuations. The study investigates the unsteady processes affecting blowing snow particle energy balances to better understand the form of advection and entrainment correction terms for sublimation calculations. To this end, a scaling relationship previously applied to near-neutral atmospheric surface layer data is tested to

represent turbulent event frequency as a function of Variable Interval Time Averaging (VITA) parameters. Data used to validate this model consist of field measurements of three-dimensional wind velocities and sonic temperatures during blowing snow events at the blowing snow study site in the Fortress Mountain Snow Laboratory (FMSL), Canadian Rockies (Figure 1). These data are supplemented by observations of nearby temperature, relative humidity, and wind speeds at three additional meteorological stations within FMSL. This provides a broader environmental context in which to understand potential thermodynamic feedback mechanisms beyond the blowing snow study site scale. The scaling relationship also gives a real-

world context for recent numerical studies on the impacts of non-stationarity on blowing snow sublimation rates.

## 2 Methods

### 2.1 Field Data

Fortress Mountain Snow Laboratory (FMSL) is located in the Kananaskis Valley in the Canadian Rockies of southwestern
Alberta, Canada. FMSL is surrounded by very complex terrain, with multiple nearby 2900m peaks having >100m vertical rock faces. The blowing snow study site is situated on a plateau at 2000m at the base of a closed ski resort, providing ample upwind fetch with minimal obstruction from trees or buildings (Figure 1 inset). Winter air temperatures at the FMSL blowing snow site typically range from -20° to +5°C, with frequent midwinter downslope chinook (föhn) wind events. Snow depth at the blowing snow site remains fairly constant through the midwinter at approximately 1 m, with fresh snowfall frequently
redistributed by wind events.

Ultrasonic temperature and wind velocity time series were observed at the FMSL blowing snow study site using two Campbell Scientific CSAT3 sonic anemometers sampling at 50 Hz from November 2015 to March 2016. The anemometers were positioned on the same mast at heights above the snow surface varying over 0.1-0.4 m and 1.4-1.7 m with snow surface accumulation or erosion. These anemometers will be referred to as the 20cm and 150cm anemometers throughout the
remaining text. Extensive analysis of this dataset has already provided new insights into the turbulent mechanisms for blowing snow transport [Aksamit and Pomeroy, 2016, 2017, 2018]. The turbulent structures scrutinized here have previously been coupled with Particle Tracking Velocimetry (PTV) and high-speed video analysis of Aksamit and Pomeroy [2017, 2018] to better understand the wind-snow coupling. For each night (20 Nov, 4 Dec, 2015 and 3 Feb, 3 Mar 2016), the time series spanning the entire duration of blowing snow video recording (from 18:00 local time to the end of video collection,
approximately 23:59) was divided into 15-minute intervals and analyzed. One additional night of meteorological data was analyzed to compare energy transport mechanisms under much windier conditions, even though PTV analysis was not available. This additional night, January 21, 2016 had much stronger winds, gusting up to 15 m s$^{-1}$ because of the presence of a chinook event. In this valley, these events have been previously associated with high blowing snow sublimation rates [MacDonald et al., 2018]. The mean temperatures varied from -7°C during the previous three days to +3°C during the night of
Jan 21. This resulted in a much larger difference between air and snow surface temperatures, and provided an interesting comparison of conditions that are critical for snow sublimation at short timescales [Sharma et al., 2018].

Three other FMSL stations near to the blowing snow measurement site provide complementary 15-minute relative humidity (with respect to ice), air temperature and wind speed measurements (Figure 1). As relative humidity measurements were not available at the blowing snow study site during the 2015-2016 study season, these additional stations provided downwind test
sites for evidence of the occurrence of large-scale thermodynamic feedbacks. The nearest complementary site is a sheltered forest (Powerline) station approximately 400 m away and 30 m higher in elevation [Smith et al., 2017]. Additionally, there are two exposed sites, including a ridgetop (Canadian Ridge) and lee side of ridge (Canadian Ridge North) that are both approximately 600 m downwind and 200 m higher in elevation. The Powerline station receives much less wind than the exposed sites or blowing snow site and is much less susceptible to snow redistribution. Temperature and relative humidity

from a previous study spanning January to March 2015 at the blowing snow study site showed a good correlation with the three nearby sites over a sample of 3,700 15-minute averages. $R^2$ values for air temperature between the blowing snow site and Canadian Ridge, Canadian Ridge North, and Powerline were 0.82, 0.83, and 0.97, respectively, and were 0.61, 0.62, and 0.80, for relative humidity, respectively. All correlation coefficients were statistically significant at 99.99%. Meteorological variables at the blowing snow study site can be found in Table 1. The Monin-Obukhov stability parameter, $\zeta$, was calculated following Monin [1970] and Stull [1988] such that

$$\zeta = \frac{z}{L},$$

$$L = -\frac{u_*^3 \theta_0}{\kappa g \overline{w'\theta'}}$$

where $\theta_0$ is the potential temperature at the 20 cm anemometer, and $u_* = \left(\overline{u'w'}^2 + \overline{v'w'}^2\right)^{1/4}$. We use the sonic temperature in lieu of potential temperature as suggested by Stull [1988] as there were no atmospheric pressure measurements at the study site. Turbulence intensity was calculated as

$$I = \frac{\overline{u'^2 + v'^2 + w'^2}}{(\overline{u}^2 + \overline{v}^2 + \overline{w}^2)^{1/2}}.$$

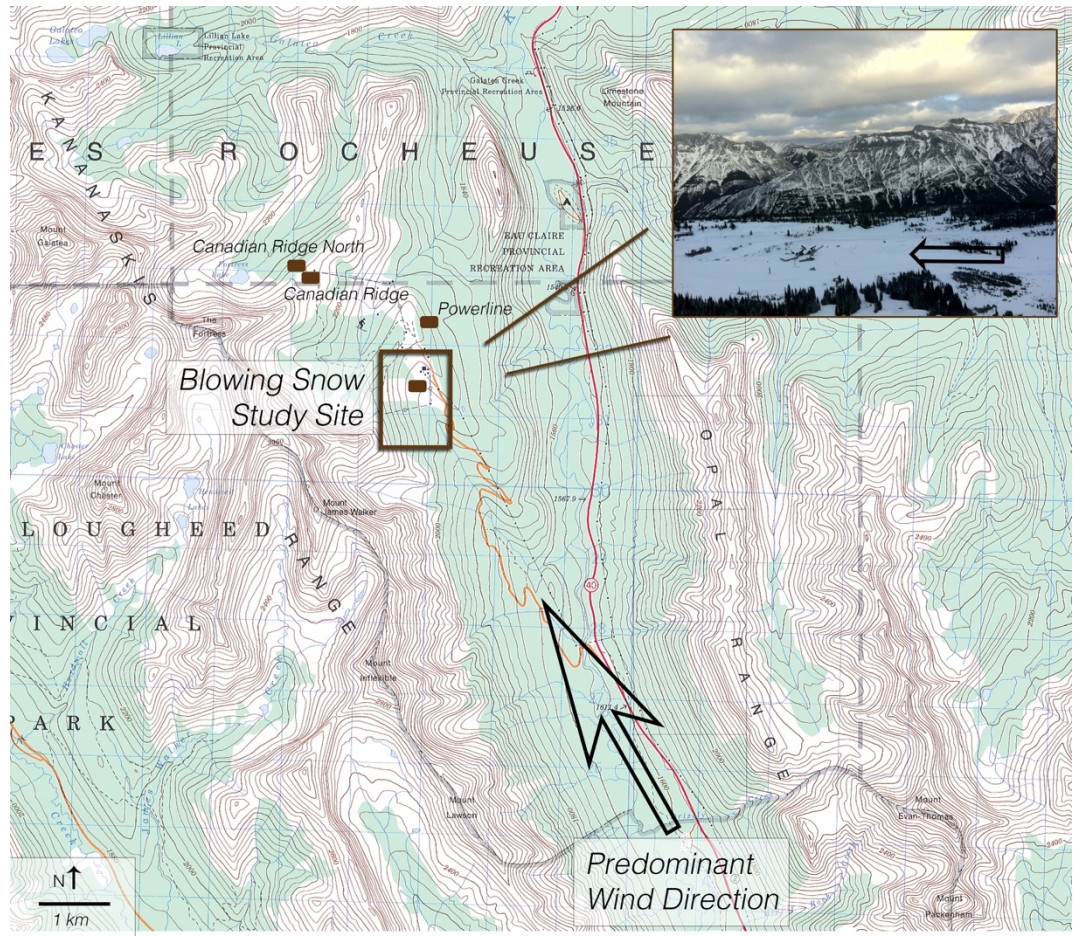

**Figure 1: Overview of blowing snow study site and adjacent micrometeorological stations at the Fortress Mountain Snow Laboratory in the Kananaskis Valley, Alberta, Canada. The prominent down-valley wind direction is noted on the map and on the site photo inset. Topographic map produced by the Canada Centre for Mapping, Natural Resources Canada, © Her Majesty The Queen in Right of Canada.**


| Date | 140 cm Wind Speed (m s⁻¹) | Monin-Obukhov (non-dim) | Air Temp 20cm, 140 cm (°C) | Snow Surface Temp (°C) | Turb. Intensity range (%) |
|---|---|---|---|---|---|
| **Nov 20, 2015** | 3.8 (1.7) | 0.06 (0.07) | -10.4 (1.06), -9.7 (0.9) | -11* | 93.3 (45.0) |
| **Dec 4, 2015** | 4.1 (1.8) | 4.6e-4 (1.7e-4) | -4.2 (0.4), -3.9 (0.4) | -4 | 72.7 (31.8) |
| **Jan 21, 2016** | 5.5 (2.2) | 6.4e-4 (2.4e-4) | 2.1 (0.7), 2.6 (0.7) | -2* | 66.8 (36.6) |
| **Feb 3, 2016** | 3.5 (1.7) | 0.02 (0.02) | -10.3 (0.5), -9.9 (0.6) | -10 | 66.6 (26.6) |
| **Mar 3, 2016** | 3.4 (1.7) | 4.0e-3 (3.1e-3) | -3.2 (0.9), -2.3 (0.7) | -5 | 94.8 (57.9) |

**Table 1: Meteorological Variables for Five Nights of Observations. *Snow surface temperature taken from the nearby Powerline meteorological station.**

## 2.2 Modified VITA Analysis

Variable Interval Time Averaging (VITA) is a method of timeseries analysis that identifies significant turbulent events as periods of high local variance. For a given timeseries f(t), the VITA method selects times such that

$$\hat{f}(t,T) = \frac{1}{T}\int_{t-T/2}^{t+T/2} f(t)^2 dt - \left[\frac{1}{T}\int_{t-\frac{T}{2}}^{t+\frac{T}{2}} f(t)dt\right]^2 > k_V\, \overline{f^2}, \tag{1}$$


where T is a statistically or experimentally determined averaging time, $k_V$ is a user-defined VITA threshold, and the overbar indicates a spatial or temporal average. For the analysis conducted here, as in a previous blowing snow investigation at this site [Aksamit and Pomeroy 2017], f(t) is taken to be the instantaneous Reynolds stress $\tau(t) = -\rho_{air}u'(t)w'(t)$, where $u'$ and $w'$ are the instantaneous fluctuations around 15-minute averages of streamwise and vertical velocities. Eq (1) needs two user-defined parameters, the temporal neighborhood T and the magnitude threshold $k_v$. To increase objectivity, and connect our turbulent events to extensively-studied and physical turbulent structures, a modified VITA analysis used here also includes a quadrant hole analysis criterion [Lu and Willmarth, 1973; Morrison et al., 1989]. Subsequent to finding a time meeting the conditions defined by Eq 1, we then identify the neighborhood surrounding that time where $\tau$ also exceeds a given threshold, $k_Q$, often called the "quadrant hole" value:


$$|\tau(t)| \geq k_Q\rho_{air}\sqrt{\overline{u'^2} + \overline{w'^2}}. \tag{2}$$

The modified VITA analysis was conducted over a range of thresholds ($0.01 \leq k_V \leq 0.05, 0.05 \leq k_Q \leq 4$) and averaging times ($0.5 \leq T \leq 40$ s) found in the boundary layer and sediment transport literature [Morrison et al. 1989; Narasimha and Kailas 1987, 1990; Bauer et al. 1998; Sterk et al. 1998; Wiggs and Weaver 2012]. This provides a relatively robust gust identification scheme that delimits significant turbulent events of varying duration and velocity magnitude. The subsequent analysis focuses on sweeps ($u' > 0, w' < 0$) and ejections ($u' < 0, w' > 0$) as they disproportionately contribute to the total surface Reynolds stress, are frequently used in models of turbulent boundary layer structures, have been identified to play crucial roles in boundary layer heat flux and aeolian transport [e.g. Bauer et al., 1998; Adrian et al., 2000; Garai and Kleissel, 2011; Aksamit and Pomeroy, 2017]. Please refer to Wallace [2016] for a recent review of the theory and experiments surrounding quadrant analysis and sweep-ejection cycling. The modified VITA algorithm categorized a turbulent event as a sweep or ejection if the parameterized curve $s(t) = \langle u'(t), w'(t)\rangle$ passes through only one of the two quadrants during the event (Q2 for ejections and Q4 for sweeps).

In this study, the concurrent sonic temperature signal response was also measured and the fluctuation from the 15-minute mean air temperature was computed to identify the presence of relatively warmer or colder air during a particular event with respect

to mean conditions. For the air temperatures during this study, CSAT3 anemometers have an error of less than $\pm 0.002°C$, which is considered negligible [Campbell Scientific, 2018]. Following Kailas and Narasimha [1994], events detected with larger thresholds are referred to as "stronger" or "more intense."

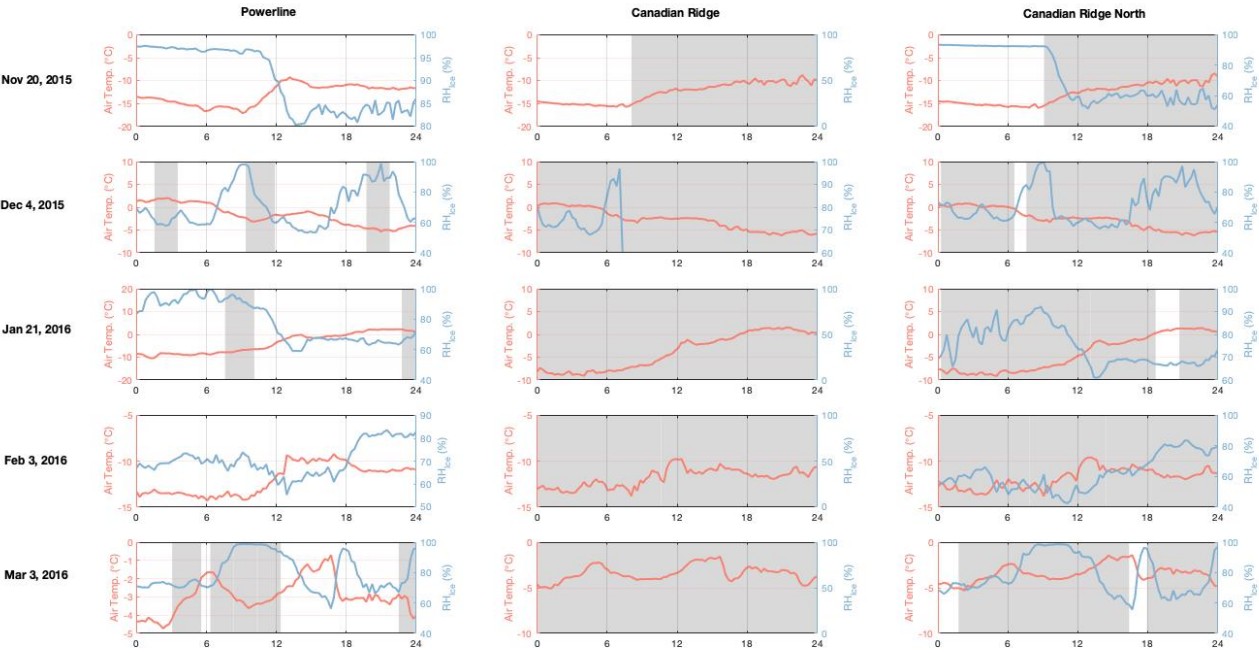

**Figure 2: 15-minute average temperature and relative humidity measurements at approximately 2 m above the snow surface during the five nights of investigation at three nearby micrometeorological stations. Flagged data have been removed from the time series**
**and presented as gaps. Shaded gray areas are times when concurrent average wind speeds were above 3 m s$^{-1}$. Note the correlation between sites for both variables, and the varying y-axes between plots.**

## 3 Results

### 3.1 Modified VITA Results

During each blowing snow storm, there was no definitive evidence of humidity saturation or thermodynamic feedback at two
of the three nearby weather stations (Figure 2). Unfortunately, RH data were unusable at the Canadian Ridge site for four of the five nights. Increases in RH were typically coupled with decreases in air temperature, and were transient in nature. The complex topography and enhanced turbulent mixing at FMSL may be responsible for this as indicated by the modified VITA analysis below. Indeed, though all three sites are situated in close proximity to each other, there is limited correlation for meteorological variables between all three, suggesting incredibly complex wind flow and energy fluxes in this alpine zone.

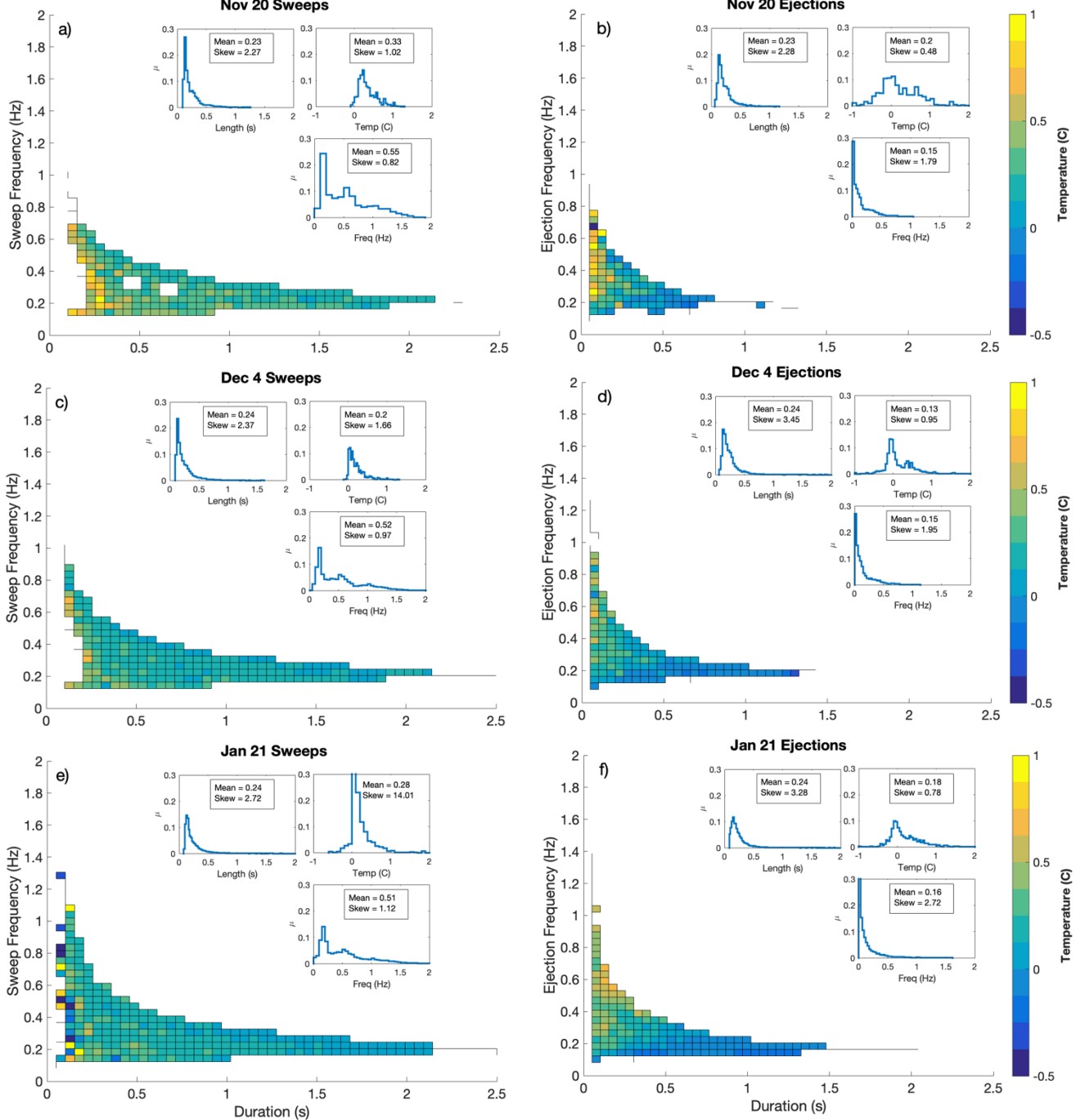


**Figure 3: Bin-averaged temperature fluctuations of near surface anemometer from the 15-minute mean for events of specific return frequency and event duration for recordings over each blowing snow storm. Insets are plots of probability distribution functions of event duration, temperature deviation and event frequency for each storm and type of event.**

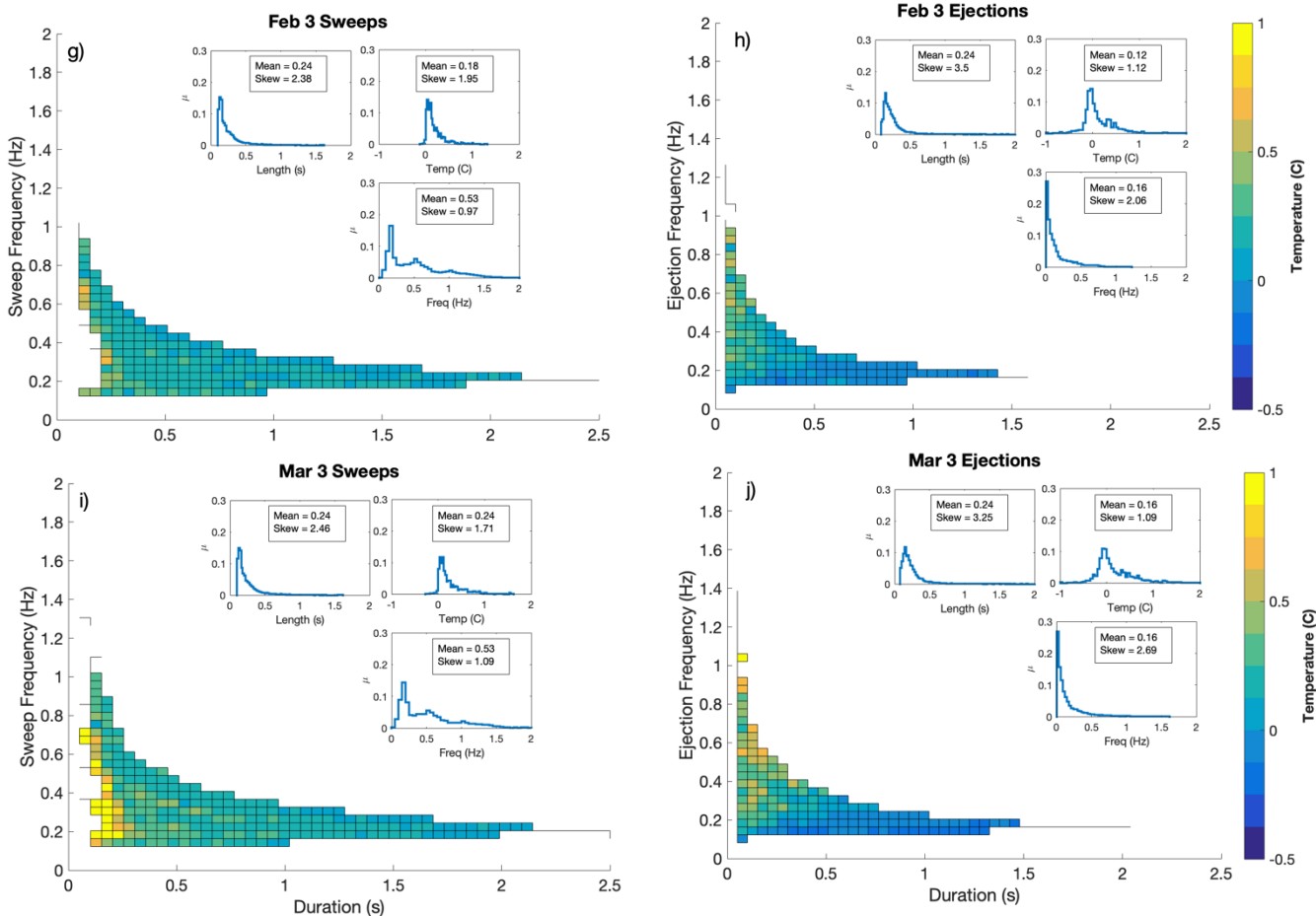

**Figure 3: continued.**

For each VITA-identified event, instantaneous temperature deviations from the 15-minute mean were computed to represent the magnitude and sign of turbulent temperature mixing with respect to slower meteorological changes over the nights. Aksamit and Pomeroy [2017] noted that there is no objective choice of averaging time or event threshold for the modified VITA analysis. As such, sonic temperatures during active turbulent events were examined over a variety of thresholds to determine a range of behaviour. The recurrence frequencies and average durations of sweep and ejection events for each threshold combination illustrated the average prevalence of sweep or ejection motions. Further sensitivity analysis of the impact of VITA parameters on wind-snow coupling has been conducted by Aksamit and Pomeroy [2017].

For each blowing snow storm, 3D point-clouds of mean recurrence frequency, event duration and event temperature deviation were calculated for the 20 cm sonic anemometer. Each point represents the values from one choice of averaging time and modified VITA thresholds for a 15-minute observation period as discussed in Section 2.2. The 3D plots contain significant overlap, so for clarity, the mean temperature deviations were averaged over small ranges of event duration and frequency, as

shown in Figure 3. Inset in each subplot are three probability distributions computed from the original point-clouds for each blowing snow storm: distributions of temperature deviation, event duration, and event frequency. Mean and skewness values are noted next to each distribution.

The analysis revealed for the four non-chinook blowing snow storms (Nov 20, 2015; Dec 4, 2015; Feb 3, 2016; Mar 3, 2016) that sweeps consistently brought warmer air to the near-surface anemometer. This can be seen as the coloured temperature plots show average temperature deviations greater than zero for nearly all event duration and frequencies over each storm. Probability distributions show very few sweep events with negative temperature deviations, as well as a consistent positive mean and skewness. The chinook storm on January 21, 2016 had a positive mean and skewness, but exhibited short cold air

bursts as well. Mean temperatures for sweeps were warmer than ejections for all blowing snow storms.

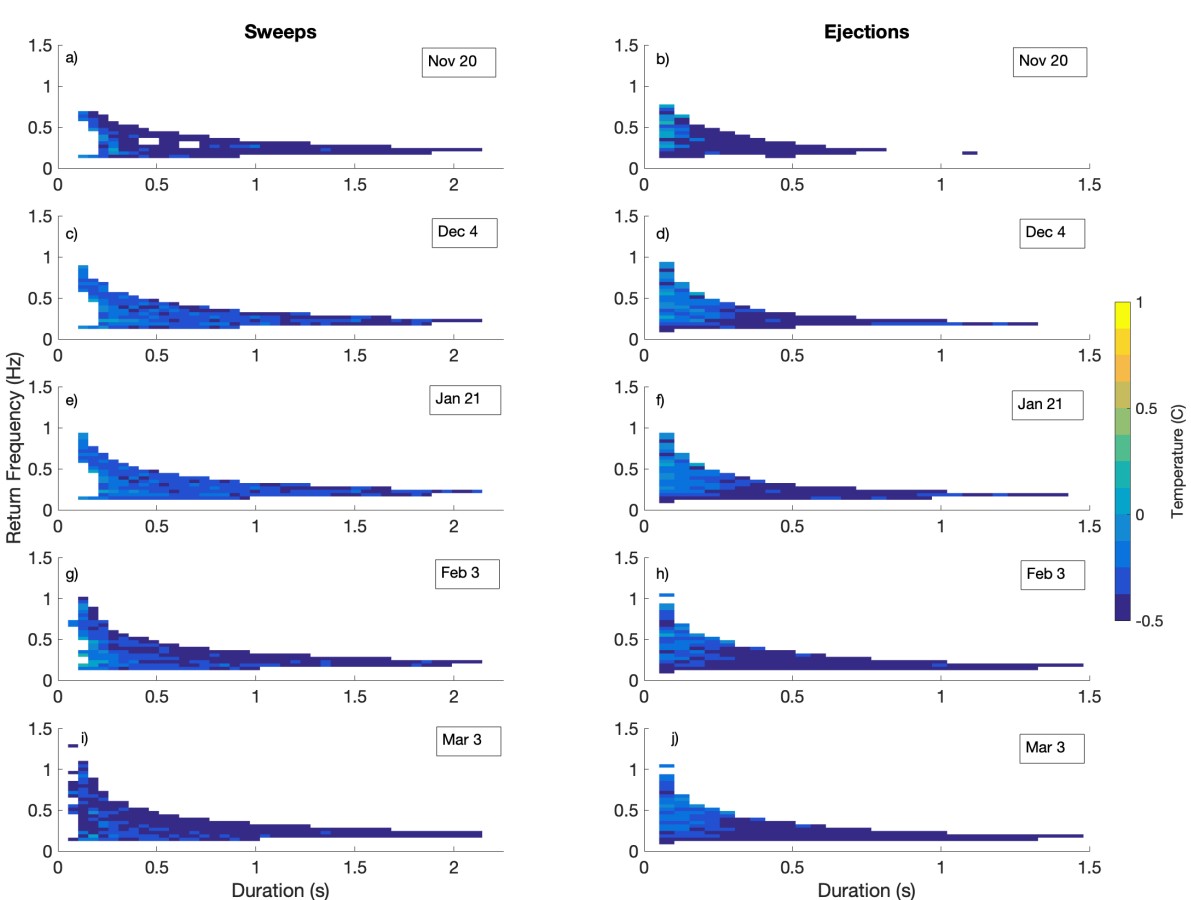

**Figure 4: The same measurements as shown in Figure 3, now subtracting the upper (140 cm above the surface) anemometer mean from the near-surface (20 cm) sweep and ejection temperatures for various blowing snow storms. Note the predominantly colder fluctuations.**

Modified VITA analysis on the 140 cm anemometer wind and temperature time series showed similar results. Interestingly, analysis comparing 20 cm anemometer event temperatures to the 140 cm anemometer means revealed near-surface sweeps often occur with colder signatures than the nearby 140 cm anemometer means (Figure 4). This is in contrast to what was found relative to surface temperatures. As these measurements were all made during the night and over a continuous snowcover with a slightly-stable temperature profile, this indicates relatively warm upper-air mixing with cold near-surface air that resulted in

a mixed temperature value between the two anemometer means. For example, blocks on the left side of Figure 3i show a group of sweeps that were 1°C warmer than the mean temperature of the 20 cm anemometer (bright yellow), but in Figure 4i, the same group of sweeps were 0.5°C colder than the 140 cm anemometer mean (dark blue). Color scales are equivalent in Figures 3 and 4. This effect is further supported by the mean anemometer temperatures detailed in Table 1.

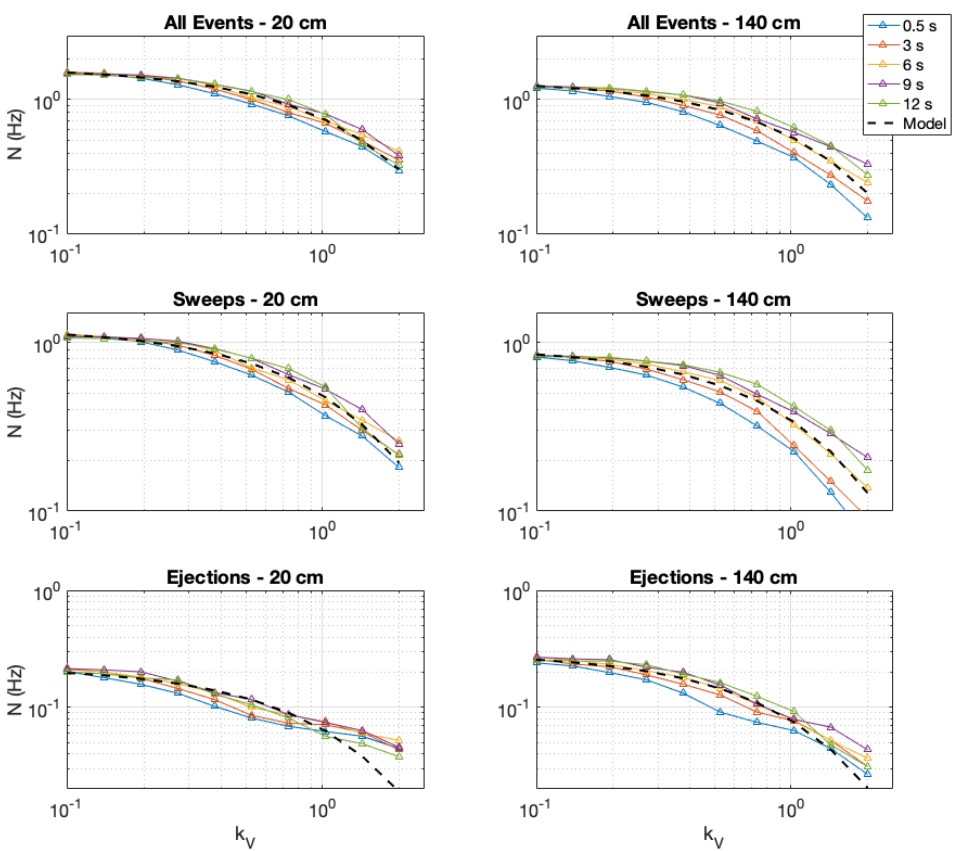

**Figure 5: Example of variation of event frequency ($N$) over VITA thresholds ($k_V$) for different averaging times ($0.5 < T < 12$ s) at**
**$k_Q = 1$ for one 15-minute study period on December 4. Least-squares fitted Eq. 1 curves are overlaid as dashed line for each collection of events: All modified VITA events, sweeps, and ejections as identified at the 20 cm (low) or 140 cm (high) anemometers.**

Ejections present a less clear story but also appear very effective for surface layer mixing. Depending on the intensity of the events detected, ejections could be either warmer or colder than the 20 cm anemometer mean (Figure 3). Over all blowing

snow storms, ejection temperatures had a lower mean and were less positively skewed. This can be explained by their physical definition of moving air vertically away from the cold snow surface. During periods of greater atmospheric stability (November 20 and March 3) there was more variability in the temperature contributions from ejections. This may indicate stable layers of varying strength were able to form and cause less uniform mixing near the snow surface. When Monin-Obukhov coefficients were closer to zero (Table 1), indicating more neutral conditions, there was less variability in ejection temperatures, indicating

a smaller range of temperatures during ejection induced mixing. This can be seen by a comparison of Table 1 values and Figs 3 and 4 probability insets. This mixing process is discussed in more detail in Section 4.

  Over all nights, sweeps were of longer duration than ejections and had a higher frequency of occurrence. The probability curves in Figure 3 show a second sweep return frequency peak around 0.5 Hz for all nights. This is not present in the ejection frequency probabilities, which only has a single low frequency peak. These sweep and ejection motions have not been

connected to a specific flow topology in these experiments (e.g. a hairpin bursting process) due to the complexity of the flow in this complex terrain. It very well may be the case that the sweep signatures are caused by both outer-layer and inner-layer motions as previously suggested by Aksamit and Pomeroy [2017]. The ejections occur less often because of the rarity of large positive $w'$ values close to the snow surface, and are thus present only during a less common generating mechanism.

### 3.2 Scaling Relation

Though several differences in the datasets exist, the near-neutral and slightly-stable conditions found during the blowing snow storms sampled suggest a Kailas and Narasimha [1994] scaling relationship may exist:

$$N = N_0 e^{-\alpha(k_V - 1)}. \tag{3}$$

Here, $N$ is the recurrence frequency of a given modified VITA turbulence event type, $N_0$ and $\alpha$ are fitting parameters with $N_0$ known as the characteristic frequency. This scaling analysis focused on the case where $k_Q = 1$ as this resulted in a good compromise between too many and too few events detected and is a standard value previously used for turbulent motion

identification at this site [Aksamit and Pomeroy, 2017]. Though the present modified VITA analysis involves an additional step in the identification algorithm as compared to the original work of Kailas and Narasimha [1994], a similar invariance (small standard deviation) in the log of the return frequency, $log(N)$, was present over varying averaging times $T$ for each VITA threshold $k_V$. This resulted in a good fit of Eq. 1 for the return frequencies of the total number of modified VITA events, as well as for sweeps and ejections individually. One example of this fitting for one 15-minute period on December 4 is shown

in Figure 5. The squared $\ell^2$-norm of the residuals for each minimized least-squares fit are presented in the document supplement, as are the characteristic frequencies, $N_0$.

| Anemometer Height | All Events | Sweeps | Ejections |
|---|---|---|---|
| 20 cm | 0.80 (0.08) | 0.49 (0.04) | 0.1 (0.03) |
| 140 cm | 0.54 (0.03) | 0.35 (0.01) | 0.09 (0.02) |

Table 2: Mean characteristic frequencies $N_0$ for turbulent events at both blowing snow site anemometers. The standard deviation of nightly means is shown in parentheses and indicates minimal changes between nights.


Total mean values (and standard deviations between nightly means) of $N_0$ are detailed in Table 2, for all turbulent events, only sweeps, and only ejections at both 20 cm and 140 cm anemometers. There was little variation of $N_0$ between nights of observation as seen in the relatively small standard deviation values. This suggests persistent flow features at this site from one night to the next that may be due to a persistent topographically induced flow feature or turbulence generating mechanism

at the study site. As could be expected from the analysis presented in Figure 3, the characteristic return frequencies ($N_0$) of all turbulent events and for sweep events were greater than those for ejections. Of particular interest in this scaling relationship is a clear difference between $N_0$ for the 140 cm and 20 cm anemometer observations for both total events and solely sweep events. Over all nights, the characteristic frequency for total events was lower at the 140 cm anemometer, which corresponded with a drop in the number of sweeps, whereas the characteristic frequency of ejections was nearly identical at both heights.

The threshold criteria in Eq. (1) and (2) varies for measurement location and time, scaling by mean values calculated over each observation period at each anemometer. This implies that there were fewer relatively-large sweeps away from the surface, and a possible shift in turbulent structure dynamics. As well, this supports the suggestion in Section 3.1 that the mechanisms generating sweeps and ejections may be different, with less common flow features resulting in the ejections.

**4 Discussion**

The same strong sweep events that have been previously found to be highly relevant for blowing snow initiation and transport at this site [Aksamit and Pomeroy, 2017], are also responsible for advecting warmer-than-average air to the near-surface layer. This is a critical insight for blowing snow sublimation modeling as the periods with greater than average blowing snow transport coincide with the presence of warmer than average air (sweeps).

Previous theoretical work has concluded that suppression of sublimation of surface and blowing snow may occur if moisture fluxes near the surface are counterbalanced solely by diffusion [Bintanja, 2001]. Dover and Mobbs [1993], Dery and Taylor [1996], Dery and Yau [1999, 2001], Groot Zwaaftink et al. [2013] and others have suggested that blowing snow sublimation could be a self-limiting process when thermodynamic feedbacks are included in a steady-state boundary layer model. However, these models did not account for warm- or dry-air entrainment, nor the temporal correlation of transport bursts with warm-air

entrainment. This missing forcing term may explain the lack of evidence of saturation in blowing snow field studies in the steppes of Russia, high plains of Wyoming (USA), prairies of Saskatchewan, alpine mountains of Alberta and arctic tundra of

the Northwest Territories (Canada), and East Antarctica [e.g. Dyunin, 1959; Schmidt, 1982; Pomeroy and Li, 2000; Musselman et al., 2015; and Grazioli et al., 2017]. The evidence of frequent regeneration of warm air near the surface through advection or entrainment processes helps explain the discrepancy with diffusion-dependent models.

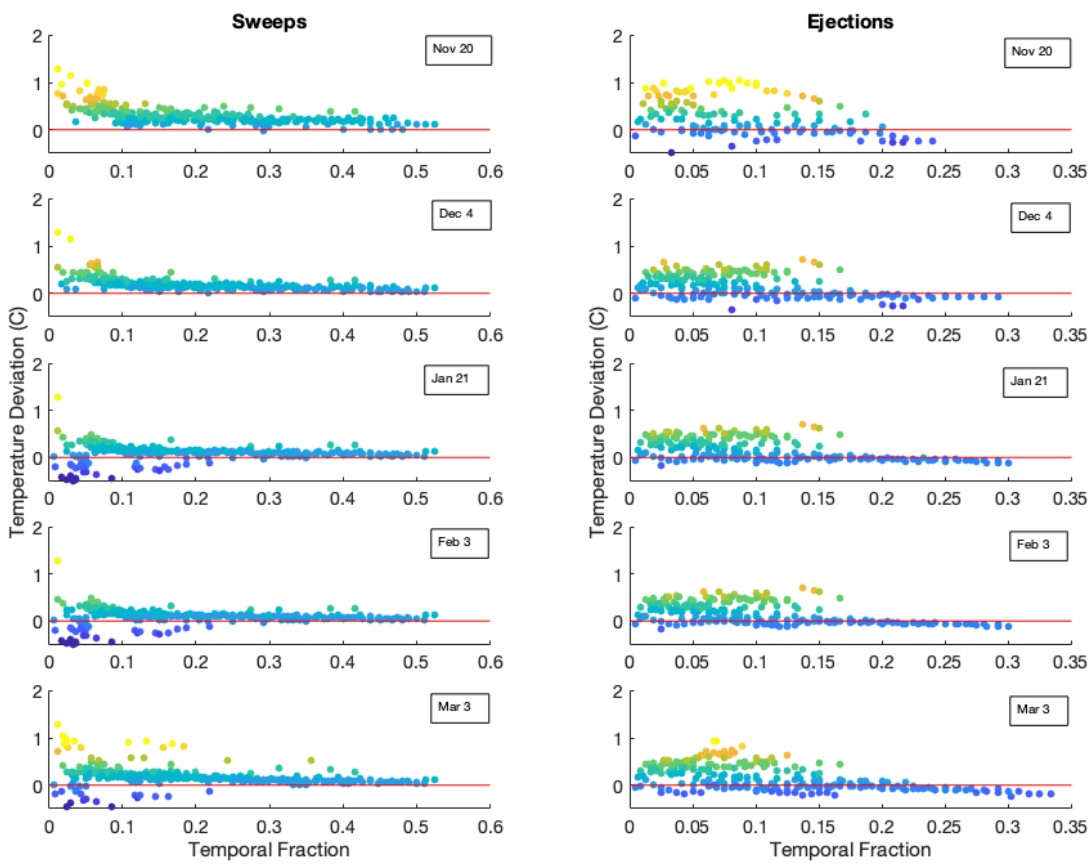

 **Figure 6: Fraction of time series occupied by sweep and ejection events of specific temperatures at the 20 cm anemometer. Refiguring of data in Figure 3 with same color scale. Colors here also correspond to y-axis values.**

Recent model simulations by Sharma et al. [2018] and Dai and Huang [2014] have shed light on the importance of temperature and wind speed fluctuations at the timescales of the sweep and ejection processes highlighted here. The comparison of the 275 Sharma et al. [2018] large-eddy-simulation-driven sublimation model with the widely used steady-state model of Thorpe and Mason [1966] revealed that transient sublimation rates approached the steady-state model only after time periods ranging from $10^{-2}$ to 10 s, depending on particle diameter and ventilation rates. At the velocities and particle sizes typical for the present study, their time to model relaxation was around 1 s. Furthermore, Dai and Huang [2014] found transient rates of sublimation in the saltation layer that reached steady-state only after 0.5-2 s. These modeled relaxation times are precisely in the range of

turbulent warming and cooling events show in Figure 3. Figure 6 redisplays the data from Figure 3 with the temporal fraction of modified VITA events calculated as the product of average event duration and frequency. For each night, one can find strong ejection events contributing air temperatures 1°C warmer than the mean for 15% of the time, and warm sweeps for up to 20% of the time.

In addition to the timescale considerations and transient regimes in blowing snow sublimation calculations, Sharma et al. [2018] found temperature fluctuations of 1°C can affect instantaneous sublimation rates by as much as 100%. Given that gusts causing temperature fluctuations of this order occur up to 35% of the time, this advected energy warrants further investigation and inclusion in future models. Fortunately, a parameterization for mechanically-explained advected energy may be possible through the simple exponential scaling relationships of Kailas and Narasimha [1994].

Over short timescales, there is a direct physical relationship between temperature profiles and temperature deviations during mixing events. This is physically intuitive if one considers the relative temperature change at a doorway when opening a door of a warm building to cold surroundings. Because of this dependence on instantaneous conditions during a mixing event, however, relationships between average temperature deviation magnitudes and long-term temperature gradients are not guaranteed. Comparing the nights of investigation, there is no monotonic relationship between increases in the average 140 cm and 20 cm sonic temperature differences and average sweep event temperature deviations. For example, on March 3, 2016 there was an average temperature difference of 0.9°C between anemometers, but the average downdraft (sweep) deviation was only +0.24°C. This is a smaller contribution than on January 21, 2016 where the air temperature difference was 0.5°C and the average sweep deviation was +0.28°C. This is almost certainly because long-time averages oversimplify the turbulent bursting process, and why eddy-covariance methods are suggested over bulk profile calculations of turbulent fluxes [Foken, 2006].

The present research has, however, suggested a simple similarity scaling of the return frequency of turbulent events of intensity $k_V$ as identified by modified VITA analysis, through the exponential relationship of Kailas and Narasimha [1994]. Such an empirical correction is compatible with the attached-eddy hypothesis [Townsend, 1976; Marusic and Monty, 2019] and other similarity-scaling models of the turbulent boundary layer if the magnitude and frequency of bursts were to be defined to scale accordingly with an increase in the size of turbulent eddies away from the surface. This scaling is evident in a decrease of characteristic frequencies of turbulent events when moving from 20 cm to 140 cm measurements (Table 2, document supplement), and a natural increase in modified VITA thresholds as the magnitude of turbulence measurements increases in Eq (1) and (2) for fixed $k_V$ and $k_Q$.

This view of boundary layer mixing provides a simple platform with which to model and investigate a gust-driven regeneration function of warm-dry air in the near-surface for blowing snow sublimation calculations. The inclusion of such a statistical recurrence model could provide an empirically defined quasi-periodic source of warm and dry air to blowing snow simulations. For example, this could be included in conservation of heat equations as a natural evolution of the constant entrainment and advection functions introduced by Bintanja [2001]. In this way, it is possible to represent the mixing of distinct parcels of air

of different temperatures through commonly studied turbulent structures. Such a recurrence model would be computationally efficient and a significant step towards a physically-based blowing snow sublimation model.

Future high temporal resolution studies of air temperature and water vapour during sustained periods of above-snow-transport-
threshold wind speeds would greatly benefit the research community. Short timescale thermodynamic feedbacks to humidity from sublimation could come from similar high frequency coupling analysis with closed path hygrometers or gas analyzers at multiple heights during blowing snow events. This would allow a more complete understanding of the advection-thermodynamic feedback balance during blowing snow storms and advance the seminal profile studies of Schmidt [1982]. As advection processes are local by nature [e.g. Harder et al., 2016], characteristic frequencies of turbulent events will vary with
location and current atmospheric conditions. The small range of values of $N_0$ measured at this site during five months of this campaign suggests common flow phenomena will possibly dominate and aid in more universal applications of entrainment modeling, at least within specific seasons.

## 5 Conclusion

During an alpine blowing snow field campaign, analysis of turbulence timeseries and sonic temperatures indicate that exceptional warm air entrainment and advection events can be associated with specific turbulent structures. Over 5 nights of investigation sweeps brought relatively warm air to the snow surface, up to 1°C warmer than average near-surface temperatures. These parcels of air may also be relatively cold compared to temperatures measured only 1.2 m above, further adding to the complexity of the physics of blowing snow sublimation. Ejections also result in strong but less consistent
temperature mixing. The current lack of understanding of advection or entrainment during snow transport may explain why the thermodynamic feedback parameterizations necessary in many blowing snow sublimation models are unphysical. In fact, field measurements of atmospheric conditions during these blowing snow events showed no evidence of significant sublimation feedbacks, let alone saturation of relative humidity. An enhanced influence of mechanical mixing in boundary layers with inhomogeneous temperature distributions, for example where there is topographically induced cold-air pooling or
flow separation, may explain why sublimation rate observations and estimates can be high and can vary from study to study. The present research indicates that including a supply of warm and dry air from different near-surface regions of the flow is a physically-accurate modeling assumption. A better representation of turbulent mixing in these regions is likely necessary for the improvement of sublimation rate estimates.

At present, further investigation of the connection of blowing snow sublimation to specific atmospheric structures would be
beneficial. Specifically, vertical profiles of high frequency temperature and humidity measurements are necessary to illuminate the impact of penetrating low frequency gusts on warm, dry-air regeneration at the surface during blowing snow sublimation in different environments. This analysis would require a closed-path style water vapour measurement as snow particles could otherwise impact the signal quality. Such an experiment could provide high-resolution temperature and complementary water

vapour measurements to more directly measure the influence of gusts on sublimation rates and begin to address discrepancies
in sublimation found in different climates. As well, longitudinal studies of heat flux in near-surface layers would provide better insight into the connection between average boundary layer profiles and the presence of turbulent events of specific magnitude, frequency, and duration.

**Author Contributions** NA and JW designed the experiment and contributed to the evaluation of the results. NA performed
the field experiment and analysis. Both authors contributed to the writing of the manuscript.

**Acknowledgements** The authors acknowledge funding from the Global Water Futures Programme, the Canada Foundation for Innovation, the Natural Sciences and Engineering Research Council of Canada, Canada Research Chairs, the Global Institute for Water Security and Alberta Agriculture and Forestry. The assistance of the Fortress Mountain Resort in logistics
is gratefully noted. This dataset is in the process of being hosted on a public server by the Global Institute for Water Security.

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
