# Peer review of "Warm Air Entrainment and Advection during Alpine Blowing Snow Events"

_The Cryosphere, 2020_

## Referee Comment (RC1) · Stephen Déry (Referee) · 29 Mar 2020

Summary: This paper presents measurements of high frequency atmospheric conditions during five blowing snow events in the fall/winter of 2015/2016 observed at the Fortress Mountain Snow Laboratory (FMSL) in southwestern Alberta, Canada. The dataset includes 3-D air temperature and wind speed fluctuations from two sonic anemometers and meteorological conditions at three proximal automatic weather stations. Blowing snow conditions are stratified into "sweep" or "ejection" events based on the combination of horizontal and vertical wind speed anomalies. The observations reveal periodic incursions of dry air from aloft that sustain or enhance blowing snow sublimation during wind transport events. Sudden increases of up to 1°C in air temperatures from dry/warm air entrainment suggests increased available thermal energy for

the sublimation process. As such, dry air advection can suppress the thermodynamic feedback associated with blowing snow sublimation.

This is an interesting effort that will be of interest to the readership of The Cryosphere. The paper is very well-written and figures are generally clear and entirely appropriate to illustrate key points. My general and specific/technical comments on the paper are as follows:

General Comments:

1) The abstract lacks key information such as the period of study and the specific study site (Fortress Mountain in southwestern Alberta, Canada).

2) Section 2.1 should provide a short description of the study area and its climate. Provide the coordinates and elevation of the blowing snow study site, some information on the local topography and climate to provide the reader some geographical context.

3) Are ultrasonic humidity measurements also available at the FMSL during this field campaign? If so, it would be quite interesting to see if blowing snow sublimation, and hence humidity, responds to the rapid air temperature and wind speed fluctuations during blowing snow events. In any case, if both ultrasonic air temperature and wind speed data are available during the five blowing snow events, why not plot the corresponding sensible heat fluxes observed along with the meteorological data shown in Figure 2? At the very least, Figure 2 should include the corresponding wind speed data for all three sites.

4) At no point does the text specify whether the relative humidity data recorded at the three other FMSL stations are with respect to water or to ice. Standard meteorological instruments usually provide the former, and so if this is the case, the relative humidity data must be converted to respect to ice to make any claims or conclusions about the absence of saturation during these five blowing snow events. It should also be clear that the Powerline site is sheltered by trees and hence does not likely experience blowing snow and may not reveal evidence of thermodynamic feedbacks from its sublimation.

5) Table 1 (mis-labelled as Table 2) provides data on Monin-Obhukhov lengths but it is not clear how these are derived. Similarly the definition for turbulence intensity reported in this table is not defined in the text.

6) Blowing snow conditions are stratified into "sweep" events (u' > 0, w' < 0) and "ejection" events (u' < 0, w' > 0); yet there may also be blowing snow conditions when the horizontal and vertical wind speed anomalies are both of the same sign; therefore it is unclear why observations are provided only for the sweep and ejection events.

7) Further to this, is an assumption made that blowing snow particles have no inertia and respond instantaneously to wind speed fluctuations?

Specific Comments:

1) P. 1, line 19: Fix the language in "model modeled described provides".

2) P. 1, line 27: Snow at the surface is often subjected to transport by wind only in relatively open and windy areas; areas such as the boreal forest and taiga are much less prone to wind transport of snow. The statement here should not be so general given blowing snow is not important component of the water budget in all areas experiencing snow.

3) P. 2, lines 48-50: Some prior studies (e.g. Grazioli et al. 2017; Déry and Yau 2001) have explored turbulent mixing and entrainment of dry air into the atmospheric boundary layer with impacts to the blowing snow sublimation and should be cited here.

4) P. 2, line 54: Delete "in order".

5) P. 2, line 59: Insert "air" before "temperature". Are the relative humidity data with respect to water or to ice?

6) P. 3, line 64: Perhaps this subsection could be titled "Field data"?

7) P. 3, line 75: How strong were the winds during the chinook event on January 21, 2016?

8) P. 3, lines 76 and 77: The degree symbol is missing in the air temperature values reported here and elsewhere in the paper.

9) P. 3, line 83: Rather than "protected" use "sheltered".

10) P. 3, line 84: Replace "include" by "including".

11) P. 3, lines 87-89: At what temporal scales of the meteorological measurements are these coefficients of determination valid for? What are the associated probability values and sample numbers for each?

12) P. 4, Figure 1 and caption: Should the arrow on the map indicate the "Predominant" wind direction?

13) P. 4, Table 1: Note that this table is reported as "Table 2" but it should instead be "Table 1". Under "Date", the years for the events should also be reported. There is disparate information provided for the meteorological data, namely the range for wind speeds and Monin-Obhukov lengths and means for air temperatures. It would be more useful to have mean values and corresponding standard deviations for all events. What do the "lower" and "upper" air temperature measurements mean? At what depth are the snow temperature measurements collected? Why not report one decimal value for the snow temperature measurements in a similar fashion as to the air temperatures? Apart from these meteorological variables, why not report the mean and standard deviation in relative humidity with respect to ice?

14) P. 5, line 108: What does the subscript "v" denote in "kv"?

15) P. 5, line 110: Move "criterion" to just after "analysis". Insert a comma after "1989]".

16) P. 5, line 113: In Equation (2), is a negative sign needed before "air" given the absolute value of this quantity is taken?

17) P. 5, lines 113 and 114: Equation (2) has a term v' but the next line refers to w'. What does the subscript "Q" refer to in "kQ"?

18) P. 6, line 128: Are the relative humidity data discussed here with respect to water or to ice? Standard meteorological instruments provide the former and so should be converted to respect an ice surface to establish whether saturation is indeed achieved, or not, during blowing snow events in subfreezing conditions.

19) P. 6, Figure 2: The color legend on the bottom right of the plot shows the air temperature in blue and the relative humidity in red; yet the tick labels on the y-axes show air temperature in red and the relative humidity in blue. As such it is not possible to interpret this plot. It would also be useful for interpretation of the meteorological time series to know when blowing snow was occurring during the 5 events shown here, perhaps as grey shading on the plots.

20) P. 6, line 135: Again, specify if the relative humidity measurements are with respect to water or ice. At what temporal frequency are these data presented and at what measurement height? Why not add the corresponding wind speed data here? In the caption, change the text to "Flagged data have" and perhaps add a note that the y-axis scales vary between panels. The caption also states that there is limited correlation between sites for both variables yet on p. 3, line 88 it was reported there was high coefficients of determination for air temperature with lesser values for relative humidity.

21) P. 7, Figure 3: On the y-axis labels, spell out "Temperature".

22) P. 9, line 158: Add the corresponding years for the events.

23) P. 9, Figure 4: On the y-axis label, spell out "Temperature".

24) P. 10, Figure 5: A color legend is missing from this plot and so the results cannot be interpreted.

25) P. 11, lines 190-191: Delete "It is interesting to note that" and start the sentence with "The probability".

26) P. 11, lines 201 and 203: Equation (3) includes a "KV" term but on line 203 the text refers to "KQ". Note also the text includes both upper case and lower case letters for these subscripts.

27) P. 12, lines 211-212: What do all the subscripts used here mean?

28) P. 12, lines 212-213: Fix the language in "common characteristic topographically induced flow."

29) 12, lines 228-229: It should be clear that these statements apply to the study site only and cannot necessarily be generalized.

30) P. 12, line 231: Replace the semi-colon by a comma after "[1993]".

31) P. 12, line 234: Again, it might be useful to refer to prior studies such as Déry and Yau (2001) and Grazioli et al. (2017) that have considered turbulent mixing and dry/warm air entrainment effects on blowing snow sublimation.

32) P. 12, line 235: Did all of these studies report humidity values with respect to ice saturation or with respect to water?

33) P. 13, line 240: It is unclear what the statement "and thermodynamic feedback may require unphysical saturation bounds to be enforced" means. The Déry and Yau (1999) study imposed air at saturation with respect to ice at a lower boundary condition (at the surface) in their numerical model, a valid assumption over a snowpack. Please clarify this statement and how it relates to the present results.

34) P. 13, line 250: Write as "1 s".

35) P. 14, lines 270-272: Again, it is unclear if this statement is accurate given it is not known if the reported relative humidities are with respect to water or to ice. In any case, it is quite possible that the Fortress Mountain Snow Laboratory site is prone to downsloping winds aligned with the valley setting, thus leading to adiabatic warming and dry air intrusions near the surface. This may not be representative of other sites,

however, that experience blowing snow and so the results must be interpreted with caution as they may not be generalizable to other sites.

36) P. 15, line 300: This should read "Canada Foundation".

37) P. 15, line 311: Note the extra spaces in "effect".

38) P. 16, line 320: Insert the article # 4679 here.

39) P. 16, line 340: Add the volume and page numbers for this reference.

40) P. 17, line 352: Is the number in parentheses "(12)" the volume number? If so, then remove the parentheses.

References:

Déry, S. J. and Yau, M. K.: Simulation of an Arctic ground blizzard using a coupled blowing snow-atmosphere model, Journal of Hydrometeorology, 2, 579-598, 2001.

Grazioli, J., Madeleine, J.-B., Gallée, H., Forbes, R. M., Genthon, C., Krinner, G., and Berne, A.: Katabatic winds diminish precipitation contribution to the Antarctic ice mass balance, Proceedings on the National Academy of Sciences, 114(41), 10858-10863, 2017.

---

## Referee Comment (RC2) · Graham Sexstone (Referee) · 25 May 2020

I have reviewed the paper "Dry-Air Entrainment and Advection during Alpine Blowing Snow Events" by Aksamit and Pomeroy for publication in The Cryosphere. The study presents results from a field-based experiment at a study site in the Canadian Rockies where ultrasonic temperature and wind data at two measurement heights were collected at 50 Hz during 5 nighttime blowing snow events over one winter period. Turbulent motions during the blowing snow events were identified based on horizontal and vertical wind speed deviations and evaluated based on associated high-resolution air temperature fluctuations. Results highlight that turbulent motions during blowing snow events were responsible for temperature fluctuations by as much as 1°C suggesting that warm air advection during blowing snow processes can be an important energy

balance component and needs to be considered for improved blowing snow sublimation modeling.

I believe this paper addresses an important topic and clearly demonstrates a contribution to the field that is relevant to the readership of The Cryosphere. The paper is clearly organized and well written overall. My comments on the paper are outlined below:

General comment:

1) While this paper is generally well written, it is sometimes missing adequate detail and definition needed for the reader to adequately understand what was done. I would like to encourage the authors to go through the manuscript and provide more relevant background material and methodological details/definitions where needed. This is especially the case in the abstract section. I've outlined some areas that need more detail in my specific comments below. Although the authors have published many papers utilizing this dataset, this paper needs to be stand alone and the reader should not need to have read these previous publications in order to understand the details relevant to the current study. The length of this manuscript is rather short so expanding sections where additional detail is needed should not cause any issue.

Specific comments:

1) Lines 1 - 2: Does it make more sense for the title of the paper to be "Warm-Air Entrainment and Advection during Alpine Blowing Snow Events" based on the study design?

2) Lines 12 - 15: "Atmospheric sweep and ejection motions" should be further defined here.

3) Lines 16 – 17: Define "event magnitude" on line 18.

4) Lines 19 – 20: The "recurrence model" is not well defined. Also, the use of "model modeled described" should be revised.

5) Lines 20 – 22: Again, return frequencies and event durations is not well defined here.

6) Abstract: More details about what the experiment was and where is was completed are generally needed in this section. The abstract needs to provide enough context for it to stand alone.

7) Lines 36 – 37: This sentence needs further explained/rewritten. Are you suggesting that turbulent fluxes are calculated as a snow energy balance residual? This is not the case in most physically based snow models.

8) Lines 55 – 56: Further define VITA thresholds here?

9) Lines 56 – 60: It would be helpful to more specifically call out the "Blowing snow study site" in the text here so the reader isn't confused by the other meteorological stations when first referencing Fig 1. Furthermore, I suggest saying "These data are supplemented by observations of nearby temperature, relative humidity, and wind speeds at three additional meteorological stations within FMSL. . ."

10) Lines 60 – 62: "return frequency" of what and "event magnitude" of what? Need to define these here.

11) Lines 65 - 66: Two ultrasonic sensors at which sites? Clarifying the site descriptions in the introduction will help make this clearer.

12) Lines 101 – 102: VITA and quadrant analysis thresholds are discussed here before they are introduced in the subsequent equations which is confusing upon first read.

13) Lines 114 – 115: How were the ranges in the user identified thresholds in equation 1 and 2 that were tested in this study identified and defined?

14) Lines 116 - 118: Can you comment on the turbulent conditions that are not considered as sweeps or ejections when u' and w' are of the same sign? Are those potentially important turbulent conditions that need to be evaluated and considered in subsequent

studies?

15) Lines 135 – 137; Figure 2: The colors of the y-axis scales on these plots should be revised to match the line color reflected in the figure legend (i.e. temperature y-axis scale should be blue and RH y-axis scale should be red.

16) Lines 139 – 144: Consider moving this information to methods section.

17) Lines 167 – 169; Figure 4: Can you comment further on how the influence of the stabile atmospheric conditions and colder temperature near the surface may have resulted in the greater warmer deviations at the lower anemometer? These near surface temperature gradients over a snowpack are especially pronounced at night-time as compared to daytime conditions (see Figure 3 from Sexstone et al. 2016; https://onlinelibrary.wiley.com/doi/abs/10.1002/hyp.10864). Therefore, in the absence of this steep air temperature gradient (more characteristic of daytime conditions), would we expect to see such strong temperature deviations associated with sweep and ejection motions?

18) Lines 162 – 163: Based on their frequency, is it likely that the high resolution temperature increases associated with sweep and ejection motions could be resolved in the 15-min time-averaged data?

19) Line 189: I didn't see further discussion of this mixing process in the discussion section according with this statement. It would be good to elaborate on this in the discussion.

20) Lines 243 – 244; Figure 6: Consider swapping the Ejections and Sweeps columns on this figure to be consistent with the presentation in other figures throughout the paper.

21) 260 – 262 – Can you elaborate here on how you expect including these scaling relationships would alter biases in existing blowing snow sublimation models? For example, if a simulation of blowing snow sublimation was completed with existing models

as well as using this scaling relation for warm-air advection, how would the results change?

22) 263 – Please elaborate on the important environmental conditions that should be/need to be represented in future studies to further develop understanding of warm and dry air advection during blowing snow events. Given the study was completed at one study site only, it cannot be generalized that the study results could be applied to all snow covered environments where blowing snow occurs. What are the limiting environmental conditions of the current study (e.g., blowing snow events only observed during nighttime conditions over a limited range of atmospheric stability…or only sweep and ejection motions where analyzed?) and how can these be overcome in future experiments.

23) Line 269: Conclusions section should be numbered section 5.

24) Lines 270 – 272: Leading the conclusions section with a sentence about saturation of water vapor during blowing snow events doesn't really fit with the scope of this paper since it was not a measurement directly made at the blowing snow site and only observed at auxiliary meteorological stations.

---

## Author Comment (AC1) · 20 Jun 2020

Nikolas Olson Aksamit and John Pomeroy

naksamit@ethz.ch

Thank you for your thoughtful and insightful review comments on this paper. We have endeavoured to address them and think that by addressing them the resulting manuscript is greatly improved. Reviewer comments are preceded by number. General Comments:

1) The abstract lacks key information such as the period of study and the specific study site (Fortress Mountain in southwestern Alberta, Canada).

This information has now been included in L12-16 "To determine if specific turbulent motions are responsible for warm and dry air advection during blowing snow events, quadrant analysis and Variable Interval Time Averaging was used to investigate turbulent time series from the Fortress Mountain Snow Laboratory alpine study site in the Canadian Rockies, Alberta, Canada during the winter of 2015-2016"

2) Section 2.1 should provide a short description of the study area and its climate. Provide the coordinates and elevation of the blowing snow study site, some information on the local topography and climate to provide the reader some geographical context.

The following paragraph has now been added at the beginning of Section 2.1 "Fortress Mountain Snow Laboratory (FMSL) is located in the Kananaskis Valley in the Canadian Rockies of southwestern Alberta, Canada. FMSL is surrounded by very complex terrain, with multiple nearby 2900m peaks having >100m vertical rock faces. The blowing snow study site is situated on a plateau at 2000m at the base of a closed ski resort, providing ample upwind fetch with minimal obstruction from trees or buildings (Figure 1 inset). Winter air temperatures at the FMSL blowing snow site typically range from -20ïĆř to +5ïĆřC, with frequent midwinter downslope chinook (föhn) wind events. Snow depth at the blowing snow site remains fairly constant through the midwinter at approximately 1 metre, with fresh snowfall frequently redistributed by wind events."

3) Are ultrasonic humidity measurements also available at the FMSL during this field campaign? If so, it would be quite interesting to see if blowing snow sublimation, and hence humidity, responds to the rapid air temperature and wind speed fluctuations during blowing snow events.

No, this data is not available, though, as mentioned in the conclusions, we would also be interested in analyzing these data in this context.

4) In any case, if both ultrasonic air temperature and wind speed data are available during the five blowing snow events, why not plot the corresponding sensible heat fluxes observed along with the meteorological data shown in Figure 2? At the very least, Figure 2 should include the corresponding wind speed data for all three sites.

Thank you for the suggestion. The corresponding wind speed data has been added

to a supplemental figure, as well as highlighting periods conducive to blowing snow in Figure 2. We considered including the sensible heat flux estimates but to do so is full of uncertainty and likely errors and so is problematic for the following reason. Eddy-covariance calculations rely on assumptions of horizontally homogeneous terrain, time series stationarity, and identifying the single correct physical reference frame. Given the highly non-stationary processes we are discussing, it is not possible to select standard time frames for covariance calculations and would also be difficult to ascribe much meaning to these estimates, much as our earlier work identified problems in linking snow particle transport to EC estimates of shear stress. We think that including estimates of these fluxes would increase the uncertainty of the manuscript substantially and that the methods of estimating heat fluxes during such non-stationary flows needs to be reassessed in a very fundamental way. Essentially, we think that this analysis shows fundamental problems with using EC estimates of sensible fluxes over snow as earlier identified by Helgason and Pomeroy (2012) and suggested by Harding and Pomeroy (1996).

5) At no point does the text specify whether the relative humidity data recorded at the three other FMSL stations are with respect to water or to ice. Standard meteorological instruments usually provide the former, and so if this is the case, the relative humid- ity data must be converted to respect to ice to make any claims or conclusions about the absence of saturation during these five blowing snow events.

We agree and were showing relative humidity with respect to ice.

6) It should also be clear that the Powerline site is sheltered by trees and hence does not likely experiment blowing snow and may not reveal evidence of thermodynamic feedbacks from its sublimation.

Thank you. We mostly agree, though the patchy nature of the forest means that air masses measured in the clearing would be influenced by blowing snow. The other two sites are fully in the blowing snow flow zone. The introduction of the Powerline site

has been changed as follows: "The nearest complementary site is a sheltered forest (Powerline) station approximately 400 m away and 30 m higher in elevation [Smith et al., 2017]. Additionally, there are two exposed sites, including a ridgetop (Canadian Ridge) and lee side of ridge (Canadian Ridge North) that are both approximately 600 m downwind and 200 m higher in elevation. The Powerline station receives much less wind than the exposed sites or the blowing snow site and is much less susceptible to snow redistribution."

7) Table 1 (mis-labelled as Table 2)

Corrected

8) provides data on Monin-Obhukhov lengths but it is not clear how these are derived. Similarly the definition for turbulence intensity reported in this table is not defined in the text.

These two variables have now been mathematically defined with citations in the text.

9) Blowing snow conditions are stratified into "sweep" events ($u' > 0$, $w' < 0$) and "ejection" events ($u' < 0$, $w' > 0$); yet there may also be blowing snow conditions when the horizontal and vertical wind speed anomalies are both of the same sign; therefore it is unclear why observations are provided only for the sweep and ejection events.

There are two reasons why we have focused on sweep and ejections motions. The primary reason is that they disproportionately contribute to Reynolds stress in turbulent boundary layers. When calculating friction velocity or other turbulence metrics that utilize velocity fluctuations, these motions have a significant influence. Second, this influence is connected to a history of coherent feature. identification in turbulence. Sweeps and ejections have been associated with specific bursting mechanisms, hairpin packet structures, and other theories of boundary layer flows. Outward and inward interactions (when $u'$ and $w'$ are of the same sign) do not make the same, nor have the same connections been made to vortical features in boundary layer flows. The

following sentences have been added to the text "The subsequent analysis focuses on sweeps (u'>0,w'<0) and ejections (u'<0,w'>0) as they disproportionately contribute to the total surface Reynolds stress, are frequently used in models of turbulent boundary layer structures, have been identified to play crucial roles in boundary layer heat flux and aeolian transport [e.g. Bauer et al., 1998; Adrian et al., 2000; Garai and Kleissel, 2011; Aksamit and Pomeroy, 2017]. Please refer to Wallace [2016] for a recent review of the theory and experiments surrounding quadrant analysis and sweep-ejection cycling."

10) Further to this, is an assumption made that blowing snow particles have no inertia and respond instantaneously to wind speed fluctuations?

No, this assumption is not made in the text, nor is it necessary for our results. It is unclear to the authors how this conclusion was made by the reviewer. Specific Comments:

11) P. 1, line 19: Fix the language in "model modeled described provides".

This line has been changed to "The recurrence model described herein provides a significant step towards a more physically-based blowing snow sublimation model"

12) P. 1, line 27: Snow at the surface is often subjected to transport by wind only in relatively open and windy areas; areas such as the boreal forest and taiga are much less prone to wind transport of snow. The statement here should not be so general given blowing snow is not important component of the water budget in all areas experiencing snow.

Thank you – very true. We have clarified these statements as the following: "However, after snow has fallen, it is often subjected to sublimation while at rest or amplified in-transit sublimation during redistribution. Blowing snow redistribution can result in vast amounts of frozen water moving between basins or, in the case of sublimation, being removed entirely from the surface water budget in wind swept regions."

13) P. 2, lines 48-50: Some prior studies (e.g. Grazioli et al. 2017; Déry and Yau 2001) have explored turbulent mixing and entrainment of dry air into the atmospheric boundary layer with impacts to the blowing snow sublimation and should be cited here.

Thank you. These citations have now been mentioned and included.

14) P. 2, line 54: Delete "in order".

Corrected.

15) P. 2, line 59: Insert "air" before "temperature".

Inserted.

16) Are the relative humidity data with respect to water or to ice?

This has been clarified and is presented with respect to ice.

17) P. 3, line 64: Perhaps this subsection could be titled "Field data"?

This title has been changed accordingly.

18) P. 3, line 75: How strong were the winds during the chinook event on January 21, 2016?

This line has been changed as follows: "This additional night, January 21, 2016 had much stronger winds, gusting up to 15 m s-1 because of the presence of a chinook event."

19) P. 3, lines 76 and 77: The degree symbol is missing in the air temperature values reported here and elsewhere in the paper.

This has been corrected throughout the text

20) P. 3, line 83: Rather than "protected" use "sheltered".

This has been changed.
21) P. 3, line 84: Replace "include" by "including".

This has been changed.

22) P. 3, lines 87-89: At what temporal scales of the meteorological measurements are these coefficients of determination valid for? What are the associated probability values and sample numbers for each?

These values have been added to the text.

23) P. 4, Figure 1 and caption: Should the arrow on the map indicate the "Predominant" wind direction?

This has been changed. Thank you.

24) P. 4, Table 1: Note that this table is reported as "Table 2" but it should instead be "Table 1".

This has been changed.

25) Under "Date", the years for the events should also be reported.

This has now been changed. Thank you.

26) There is disparate information provided for the meteorological data, namely the range for wind speeds and Monin-Obhukov lengths and means for air temperatures. It would be more useful to have mean values and corresponding standard deviations for all events.

Thank you for the suggestion. Table 1 has been updated accordingly.

27) What do the "lower" and "upper" air temperature measurements mean?

We have clarified that we are referring to specific anemometers. Throughout the text, we have replaced the "upper" and "lower" designations with numeric 140 cm and 20 cm heights to differentiate between the two anemometers.

28) At what depth are the snow temperature measurements collected? Why not report one decimal value for the snow temperature measurements in a similar fashion as to the air temperatures?

These are snow surface measurements made in the first mm of the snow surface. These measurements were made manually and reported in a field notebook on the days of the experiment. Unfortunately, the temperatures were not recorded to the first decimal place so that precision did not exceed the accuracy of the thermometers used.

29) Apart from these meteorological variables, why not report the mean and standard deviation in relative humidity with respect to ice?

The formatting of the table has been changed in accordance with this suggestion.

30) P. 5, line 108: What does the subscript "v" denote in "kv"?

We have clarified that k_v refers to a user defined threshold in the VITA equation.

31) P. 5, line 110: Move "criterion" to just after "analysis". Insert a comma after "1989]".

This has been changed.

32) P. 5, line 113: In Equation (2), is a negative sign needed before "air" given the absolute value of this quantity is taken?

Equation 2 (and much of section 2.2) has been now been rewritten for clarity. This redundancy is no longer present in the text.

33) P. 5, lines 113 and 114: Equation (2) has a term v' but the next line refers to w'.

This inconsistency has now been corrected.

34) What does the subscript "Q" refer to in "kQ"?

We have clarified that k_Q refers to a user defined Quadrant threshold event identification algorithm.

35) P. 6, line 128: Are the relative humidity data discussed here with respect to water or to ice? Standard meteorological instruments provide the former and so should be converted to respect an ice surface to establish whether saturation is indeed achieved, or not, during blowing snow events in subfreezing conditions.

These measurements are made with respect to ice and this has been clarified in the text.

36) P. 6, Figure 2: The color legend on the bottom right of the plot shows the air temperature in blue and the relative humidity in red; yet the tick labels on the y-axes show air temperature in red and the relative humidity in blue. As such it is not possible to interpret this plot. It would also be useful for interpretation of the meteorological time series to know when blowing snow was occurring during the 5 events shown here, perhaps as grey shading on the plots.

Thank you for noting this discrepancy. The coloring on these plots has been corrected. There were no blowing snow particle detectors at these stations so it is not possible to definitively say exactly when blowing snow was present. We highlighted times when the 15-minute average windspeed was greater than 3 m/s, a threshold for transport that has been noted at blowing snow study site previously. We have also included complementary time series of snow depth measurements and wind speeds in the document supplement, but a definitive highlighting of events is not possible.

37) P. 6, line 135: Again, specify if the relative humidity measurements are with respect to water or ice.

This has been clarified when the data is introduced on line 133.

38) At what temporal frequency are these data presented and at what measurement height?

It has been added to the caption that these are 15 minute average values at approximately 2 m heights.

39) Why not add the corresponding wind speed data here?

This has been included in the updated document supplement. Relevant blowing snow transport threshold information has now been included in the updated figure.

40) In the caption, change the text to "Flagged data have" and perhaps add a note that the y-axis scales vary between panels.

This has been changed.

41) The caption also states that there is limited correlation between sites for both variables yet on p. 3, line 88 it was reported there was high coefficients of determination for air temperature with lesser values for relative humidity.

This has been clarified. There is a high correlation.

42) P. 7, Figure 3: On the y-axis labels, spell out "Temperature".

This has been changed.

43) P. 9, line 158: Add the corresponding years for the events.

This has been added. Thank you.

44) P. 9, Figure 4: On the y-axis label, spell out "Temperature".

Changed.

45) P. 10, Figure 5: A color legend is missing from this plot and so the results cannot be interpreted.

This has been added.

46) P. 11, lines 190-191: Delete "It is interesting to note that" and start the sentence with "The probability".

This has been changed.

[Figure]

47) P. 11, lines 201 and 203: Equation (3) includes a "KV" term but on line 203 the text refers to "KQ". Note also the text includes both upper case and lower case letters for these subscripts.

We have corrected the inconsistency in letter case. The use of $k_V$ in equation 3 and $k_Q$ in the following paragraph is, however, correct. As both $k_V$ and $k_Q$ are necessary for the modified VITA analysis, we restricted our model to events generated for one standard value of $k_Q$ and analyzed those results.

48) P. 12, lines 211-212: What do all the subscripts used here mean?

These subscripts have been removed and the data has been moved to a new Table 2 that is much easier to interpret. Thank you for the motivation.

49) P. 12, lines 212-213: Fix the language in "common characteristic topographically induced flow."

This line has been changed to "This suggests persistent flow features at this site from one night to the next that may be due to a persistent topographically induced flow feature or turbulence generating mechanism at the study site."

50) 12, lines 228-229: It should be clear that these statements apply to the study site only and cannot necessarily be generalized.

Comments along these lines have been added to the last paragraph of the discussion and the last paragraph of the conclusions, as well as suggestions for how further investigation will reveal what of these bursting parameters can be regarded as universal.

51) P. 12, line 231: Replace the semi-colon by a comma after "[1993]".

This has been changed.

52) P. 12, line 234: Again, it might be useful to refer to prior studies such as Déry and Yau (2001) and Grazioli et al. (2017) that have considered turbulent mixing and dry/warm air entrainment effects on blowing snow sublimation.

Thank you. These additional studies has been referenced.

53) P. 12, line 235: Did all of these studies report humidity values with respect to ice saturation or with respect to water?

Thank you for bringing up this point. Unfortunately, this information is not included in all the mentioned studies.

54) P. 13, line 240: It is unclear what the statement "and thermodynamic feedback may require unphysical saturation bounds to be enforced" means. The Déry and Yau (1999) study imposed air at saturation with respect to ice at a lower boundary condition (at the surface) in their numerical model, a valid assumption over a snowpack. Please clarify this statement and how it relates to the present results.

Thank you for bringing this to our attention. This sentence has been removed as we were largely reiterating a point made at the beginning of that paragraph.

55) P. 13, line 250: Write as "1 s".

This has been changed.

56) P. 14, lines 270-272: Again, it is unclear if this statement is accurate given it is not known if the reported relative humidities are with respect to water or to ice. In any case, it is quite possible that the Fortress Mountain Snow Laboratory site is prone to downsloping winds aligned with the valley setting, thus leading to adiabatic warming and dry air intrusions near the surface. This may not be representative of other sites however, that experience blowing snow and so the results must be interpreted with caution as they may not be generalizable to other sites.

The relative humidity is with respect to ice for temperatures below zero. It has been clarified throughout the discussion and conclusions that these are not claims about behaviour in all boundary layers, and all turbulence phenomena are local in nature. The site receives predominately upslope flows from the valley bottom during most of the blowing snow events described here.

57) P. 15, line 300: This should read "Canada Foundation".

This has been changed.

58) P. 15, line 311: Note the extra spaces in "effect".

This has been changed.

59) P. 16, line 320: Insert the article # 4679 here.

This has been added.

60) P. 16, line 340: Add the volume and page numbers for this reference.

This has been changed.

61) P. 17, line 352: Is the number in parentheses "(12)" the volume number? If so, then remove the parentheses.

This has been changed.

---

## Author Comment (AC2) · 20 Jun 2020

Thank you for this thoughtful and detailed review. We have worked to address the reviewer's comments in the updated manuscript which has benefited significantly from the edits. General comment:

1) While this paper is generally well written, it is sometimes missing adequate detail and definition needed for the reader to adequately understand what was done. I would like to encourage the authors to go through the manuscript and provide more relevant background material and methodological details/definitions where needed. This is especially the case in the abstract section. I've outlined some areas that need more detail in my specific comments below. Although the authors have published many papers utilizing this dataset, this paper needs to be stand alone and the reader should not need to have read these previous publications in order to understand the details relevant to the current study. The length of this manuscript is rather short so expanding sections where additional detail is needed should not cause any issue.

Thank you for this suggestion. The manuscript has now been expanded in the results, discussions, and conclusions sections as detailed below. Specific comments:

2) Lines 1 - 2: Does it make more sense for the title of the paper to be "Warm-Air Entrainment and Advection during Alpine Blowing Snow Events" based on the study design?

That is a very good point. We have changed that.

3) Lines 12 - 15: "Atmospheric sweep and ejection motions" should be further defined here.

In order to avoid technical language in the abstract, this sentence has now been changed to "To determine if specific turbulent motions are responsible for warm and dry air advection during blowing snow events, quadrant analysis and Variable Interval Time Averaging was used to investigate turbulent time series from the Fortress Mountain Snow Laboratory alpine study site in the Canadian Rockies, Alberta, Canada during the winter of 2015-2016."

4) Lines 16 – 17: Define "event magnitude" on line 18.

This sentence has been changed to "A simple scaling relationship was derived that related the frequency of dominant downdraft and updraft events to their duration and local variance."

5) Lines 19 – 20: The "recurrence model" is not well defined. Also, the use of "model modeled described" should be revised.

This sentence has been changed as follows: "The downdraft and updraft scaling relationship described herein provides a significant step towards a more physically based blowing snow sublimation model with more realistic mixing of atmospheric heat."

6) Lines 20 – 22: Again, return frequencies and event durations is not well defined here.

This phrasing has been removed.

7) Abstract: More details about what the experiment was and where is was completed are generally needed in this section. The abstract needs to provide enough context for it to stand alone.

Thank you for this suggestion. We have now clarified the location of the study site, what kind of data we analyzed, and which methods were used to determine our conclusions.

8) Lines 36 – 37: This sentence needs further explained/rewritten. Are you suggesting that turbulent fluxes are calculated as a snow energy balance residual? This is not the case in most physically based snow models.

Thank you, this has been clarified. While physically-based blowing snow models often include terms for turbulent flux contributions, the energy balance is never closed, and these residuals are typically attributed to different processes that were imperfectly calculated. The exact contribution of latent heat is poorly understood, especially in this environment, as the true blowing snow sublimation contribution is often only seen as the piece that is missing from the final balance. We have clarified our phrasing in the text as follows. "To accurately calculate all contributions to boundary layer energy balances, latent heat flux estimates rely on an accurate sublimation model, and a precise understanding of how much energy is available for snow particle phase change."

9) Lines 55 – 56: Further define VITA thresholds here?

We have changed the phrasing to "VITA parameters" and included a more thorough explanation of the VITA analysis in section 2.2

10) Lines 56 – 60: It would be helpful to more specifically call out the "Blowing snow study site" in the text here so the reader isn't confused by the other meteorological stations when first referencing Fig 1. Furthermore, I suggest saying "These data are supplemented by observations of nearby temperature, relative humidity, and wind speeds at three additional meteorological stations within FMSL. . ."

Thank you, these sentences have been changed as follows: "Data used to validate this model consist of field measurements of three-dimensional wind velocities and sonic temperatures during blowing snow events at the blowing snow study site in the Fortress Mountain Snow Laboratory (FMSL), Canadian Rockies (Figure 1). These data are supplemented by observations of nearby temperature, relative humidity, and wind speeds at three additional meteorological stations within FMSL. This provides a boarder environmental context in which to understand potential thermodynamic feedback mechanisms beyond the blowing snow study site scale."

11) Lines 60 – 62: "return frequency" of what and "event magnitude" of what? Need to define these here.

This sentence has been changed as follows: "The scaling relationship also gives a real-world context for recent numerical studies on the impacts of non-stationarity on blowing snow sublimation rates."

12) Lines 65 - 66: Two ultrasonic sensors at which sites? Clarifying the site descriptions in the introduction will help make this clearer.

We have clarified that we are referring to measurements at the blowing snow study site.

13) Lines 101 – 102: VITA and quadrant analysis thresholds are discussed here before they are introduced in the subsequent equations which is confusing upon first read.

This has been clarified in the text. Section 2.2 has been significantly rewritten.

14) Lines 114 – 115: How were the ranges in the user identified thresholds in equation

1 and 2 that were tested in this study identified and defined?

This has been clarified in the text. Section 2.2 has been significantly rewritten.

15) Lines 116 - 118: Can you comment on the turbulent conditions that are not considered as sweeps or ejections when u' and w' are of the same sign? Are those potentially important turbulent conditions that need to be evaluated and considered in subsequent studies?

This has been clarified in the text. Section 2.2 has been significantly rewritten.

16) Lines 135 – 137; Figure 2: The colors of the y-axis scales on these plots should be revised to match the line color reflected in the figure legend (i.e. temperature y-axis scale should be blue and RH y-axis scale should be red.

This colouring has been corrected. Thank you.

17) Lines 139 – 144: Consider moving this information to methods section.

This information is originally presented in the last paragraph of the methods section, but is reiterated here for clarity for the reader.

18) Lines 167 – 169; Figure 4: Can you comment further on how the influence of the stabile atmospheric conditions and colder temperature near the surface may have resulted in the greater warmer deviations at the lower anemometer? These near surface temperature gradients over a snowpack are especially pronounced at nighttime as compared to daytime conditions (see Figure 3 from Sexstone et al. 2016; https://onlinelibrary.wiley.com/doi/abs/10.1002/hyp.10864). Therefore, in the absence of this steep air temperature gradient (more characteristic of daytime conditions), would we expect to see such strong temperature deviations associated with sweep and ejection motions?

Thank you for this interesting question. The sign and magnitude of the temperature deviations is directly related to the instantaneous gradient of air temperatures found dur-

ing any measurement. For example, research in daytime convective boundary layers has found relative cold air contributions down to warm surfaces during sweep events (Garai et al,). We have added the following comments in the discussion: "Over short timescales, there is a direct physical relationship between temperature profiles and temperature deviations during mixing events. This is physically intuitive if one considers the relative temperature change at a doorway when opening a door of a warm building to cold surroundings. Because of this dependence on instantaneous conditions during a mixing event, however, relationships between average temperature deviation magnitudes and long-term temperature gradients are much less certain. Over the nights of investigation, there is no monotonic relationship between increases in average 140 and 20 cm sonic temperature differences and average sweep event temperature deviations. For example, on March 3, 2016 there was an average temperature difference of 0.9ïĆřC between anemometers, but the average downdraft (sweep) deviation was only +0.24ïĆřC. This is a smaller contribution than on January 21, 2016 where the air temperature difference was 0.5ïĆřC and the average sweep deviation was +0.28ïĆřC. That is, one should exercise caution if attempting to downscale long-term statistics to represent these purely local surface processes."

19) Lines 162 – 163: Based on their frequency, is it likely that the high resolution temperature increases associated with sweep and ejection motions could be resolved in the 15-min time-averaged data?

This question is unclear to the authors. The temperature deviations are measured as deviations from the 15-minute mean. Therefore, there would be no deviation if we changed the resolution of the deviations to match the mean. One would absolutely find evidence of these temperature fluctuations if looking at the 15-minute standard deviations associated with 15-minute mean data.

20) ine 189: I didn't see further discussion of this mixing process in the discussion section according with this statement. It would be good to elaborate on this in the discussion.

[Figure]

An additional three paragraphs have been added to the discussion.

21) Lines 243 – 244; Figure 6: Consider swapping the Ejections and Sweeps columns on this figure to be consistent with the presentation in other figures throughout the paper.

Thank you, this has been corrected.

22) 260 – 262 – Can you elaborate here on how you expect including these scaling relationships would alter biases in existing blowing snow sublimation models? For example, if a simulation of blowing snow sublimation was completed with existing models as well as using this scaling relation for warm-air advection, how would the results change? 22b) 263 – Please elaborate on the important environmental conditions that should be/need to be represented in future studies to further develop understanding of warm and dry air advection during blowing snow events. Given the study was completed at one study site only, it cannot be generalized that the study results could be applied to all snow covered environments where blowing snow occurs. What are the limiting environ- mental conditions of the current study (e.g., blowing snow events only observed during nighttime conditions over a limited range of atmospheric stability. . .or only sweep and ejection motions where analyzed?) and how can these be overcome in future experiments.

Thank you for the interesting questions. The following paragraphs aimed at illuminating these topics have been added to the discussion: "Over short timescales, there is a direct physical relationship between temperature profiles and temperature deviations during mixing events. This is physically intuitive if one considers the relative temperature change at a doorway when opening a door of a warm building to cold surroundings. Because of this dependence on instantaneous conditions during a mixing event, however, relationships between average temperature deviation magnitudes and long-term temperature gradients are not guaranteed. Comparing the nights of investigation, there is no monotonic relationship between increases in average 140

and 20 cm sonic temperature differences and average sweep event temperature deviations. For example, on March 3, 2016 there was an average temperature difference of 0.9ïČřC between anemometers, but the average downdraft (sweep) deviation was only +0.24ïČřC. This is a smaller contribution than on January 21, 2016 where the air temperature difference was 0.5ïČřC and the average sweep deviation was +0.28ïČřC. This is likely because long-time averages oversimplify the turbulent bursting process, and why eddy-covariance methods are suggested over bulk profile approaches to turbulent fluxes [Foken, 2006]. However, the present research has suggested a simple similarity scaling of the return frequency of turbulent events of intensity $k_V$ as identified by modified VITA analysis, through the exponential relationship of Kailas and Narasimha [1994]. Such an empirical correction is compatible with the attached-eddy hypothesis [Taylor] and other similarity-scaling models of the turbulent boundary layer if the magnitude and frequency of bursts were defined to scale up with an increase in the size of turbulent eddies away from the surface. This scaling is evident in a decrease of characteristic frequencies of turbulent events when moving from 20 cm to 140 cm measurements (Table 2, document supplement), and a natural increase in modified VITA thresholds as the magnitude of turbulence measurements increases in Eq (1) and (2) for fixed $k_V$ and $k_Q$. This view of boundary layer mixing provides a simple platform with which to model and investigate a gust-driven regeneration function of warm-dry air in the near-surface for blowing snow sublimation calculations. The inclusion of such a statistical recurrence model could provide an empirically defined quasi-periodic source of warm and dry air to blowing snow simulations. For example, this could be included in conservation of heat equations as a natural evolution of the constant entrainment and advection functions introduced by Bintanja [2001]. In this way, it is possible to represent the mixing of distinct parcels of air of different temperatures through commonly studied turbulent structures. Such a recurrence model would be computationally efficient and a significant step towards a physically based blowing snow sublimation model. Future high temporal resolution studies of blowing snow particles, air temperature and water vapour during sustained periods of above-snow-transport-threshold

wind speeds would greatly benefit the research community. Short timescale thermody-namic feedbacks to humidity from sublimation could come from similar high frequency coupling analysis with closed path hygrometers or gas analyzers at multiple heights during blowing snow events. Fast response particle detectors could give further in-sight into relationships between atmospheric and particle motions. This would allow a more complete understanding of the advection-thermodynamic feedback balance dur-ing blowing snowstorms and advance the seminal profile studies of Schmidt [1982]. As advection processes are local by nature [e.g. Harder et al., 2016], characteristic frequencies of turbulent events will vary with location and atmospheric conditions. The small range of values of N_0 measured at this site during five months of this campaign suggests common flow phenomena will possibly dominate and aid in more universal applications of entrainment modeling, at least within specific seasons." In the con-clusions, we have also suggested a longitudinal study would be greatly beneficial for understanding the variance in parameters necessary for this simple bursting model.

23) Line 269: Conclusions section should be numbered section 5.

Corrected. Thank you.

24) Lines 270 – 272: Leading the conclusions section with a sentence about saturation of water vapor during blowing snow events doesn't really fit with the scope of this paper since it was not a measurement directly made at the blowing snow site and only observed at auxiliary meteorological stations.

Thank you. This has been moved to another section of the conclusions.

Garai, A., and J. Kleissl (2011), Air and Surface Temperature Coupling in the Convective Atmospheric Boundary Layer, J. Atmos. Sci., 68(12), 2945–2954, doi:10.1175/JAS-D-11-057.1.